# Relationship between Dietary Inflammatory Index and Postpartum Depression in Exclusively Breastfeeding Women

**DOI:** 10.3390/nu14235006

**Published:** 2022-11-25

**Authors:** Hanshuang Zou, Minghui Sun, Yan Liu, Yue Xi, Caihong Xiang, Cuiting Yong, Jiajing Liang, Jiaqi Huo, Qian Lin, Jing Deng

**Affiliations:** 1Department of Nutrition Science and Food Hygiene, Xiangya School of Public Health, Central South University, 110 Xiangya Rd, Changsha 410078, China; 2Jining First People’s Hospital, Jining 272000, China; 3Department of Child Care, Changsha Maternal and Child Health Care Hospital, 416 Chengnan East RD of Yuhua District, Changsha 410007, China; 4Department of Epidemiology, School of Public Health, Sun Yat-sen University, Guangzhou 510275, China

**Keywords:** exclusive breastfeeding, breastfeeding mother, dietary inflammatory index (DII), postpartum depression (PPD)

## Abstract

(1) Background: Research has shown that chronic inflammation can increase the risk of depression. The dietary inflammatory index (DII) is a novel measure of dietary inflammation, which has been used to investigate the relationship between diet and mental disorders in adults. However, little research has been conducted to establish an association between dietary inflammation (as measured by DII) and postpartum depression (PPD) in exclusively breastfeeding women. (2) Methods: In this cross-sectional study, 293 women who were exclusively breastfeeding for 6 months or less were enrolled. The DII scores were evaluated using semi-quantitative Food Frequency Questionnaires (FFQ), and the Edinburgh Postpartum Depression Scale (EPDS) was used to measure depression levels of breastfeeding mothers during the six months following delivery. The participants were classified by tertiles, and the possibility of DII being associated with PPD was assessed by binary regression analysis. (3) Results: The average DII score was 2.32 ± 1.08, which ranged from −1.66 to 4.19. The rate of depression was 60.1%. Adjusted for potential risk factors such as age, educational level, occupational level, number of babies, number of caregivers, social support level, and sleep quality, the results showed that the lowest DII score was associated with a lower risk of PPD than the highest score (OR tertile Q_1_ vs. 3 = 0.47, 95% CI: 0.24, 0.93, *p* = 0.030). (4) Conclusions: In exclusive breastfeeding women, the inflammatory potential of dietary intake seems to be related to depression. Interventions to improve diet quality might consider including a dietary component that aims to lower chronic systemic inflammation to prevent PPD. However, the relationship between DII and PPD among Chinese women remains to be demonstrated in a larger population.

## 1. Introduction

Exclusive breastfeeding is of great importance for the health of maternal and infants. The World Health Organization (WHO) recommends exclusive breastfeeding for the first six months of life, after which complementary foods should be added and breastfeeding should be continued until two years of age or older [1]. As the prevalence of breastfeeding increases, more women are willing to try breastfeeding. However, early termination of exclusive breastfeeding is still considered a major public health concern [2]. According to the 2013 China Health and Nutrition Survey, the exclusive breastfeeding rate was 18.6% [3], far from the WHO target (50% exclusive breastfeeding rate at 6 months) [4].

Postpartum depression (PPD) is one of the most common psychiatric complications during childbearing, of which 50% to 85% of breastfeeding mothers suffer [5]. Research shows that breastfeeding mothers’ anxiety and depression often result in breastfeeding failure [6,7]. A Brazilian study showed that the risk of interrupting breastfeeding was 1.63 times higher in mothers with depressive symptoms during the first trimester than in those without PPD symptoms [8]. The results of a cohort study in China also support this view [9]; the average duration of exclusively breastfeeding was 3.90 ± 2.33 months. Mothers with depressive symptoms were more likely to stop breastfeeding or to adopt partial breastfeeding than those without depressive symptoms. PPD not only affects the health of the mother and interrupts breastfeeding, but also hinders the cognitive and emotional development of the child. However, there is still no widely accepted explanation for the mechanisms of PPD [10].

In recent years, studies have found that the inflammatory mechanism plays an important role in the occurrence of PPD [11,12,13,14]. Mothers tend to be physically tired, find breastfeeding painful, and are sleepy while breastfeeding [15,16]. Combined with stress from economic and work sources, it causes an imbalance in the immune system of the breastfeeding mother and a significant increase in pro-inflammatory cytokines, resulting in insufficient synthesis of 5-hydroxy tryptamine (5-HT) receptors [17]. Due to the decrease in this content, the 5-HT neural circuit will be blocked and thus cause the depression [18,19]. A large number of pro-inflammatory substances can also affect the function of the hypothalamic–pituitary–adrenal axis (HPA), and lead to the disorder of the HPA axis. The long-term hyperactivation of the HPA axis will finally cause depression [20,21].

Diet is essential for the development of inflammation in the body [22,23]. Evidence suggests that food composition has a direct impact on the levels of neurotransmitters, hormones, inflammatory cytokines, and gut microbiota structure [24,25,26,27], which are critical to sleep, mood, and behavioral signaling pathways [28,29,30,31]. The diet of Chinese breastfeeding mothers shows a particular pattern under the cultural influence of how to ensure the nutrition of breast milk [32]. There are two main dietary patterns of Chinese breastfeeding women [33]; both of them are less diverse and with lower macronutrients. One pattern contains mainly red meat, whole grains, and fresh vegetables (leafy), and the other category is mainly fresh vegetables (without leaves), soy milk, bacteria, and algae. Researchers have found that a diet based on red meat, whole milk, refined grains, processed meat, sweets, and soft drinks may increase the level of inflammation [34]. Conversely, a healthy diet featuring fruits, vegetables, and whole grains is negatively correlated with inflammatory factors such as CRP and IL-6 [35,36]. Based on the characteristics of dietary patterns of breastfeeding mothers in China and the relationship between inflammation and depression, we assume that the dietary patterns of Chinese mothers during breastfeeding may create an inflammatory environment in the body, potentially increasing the risk of depression.

In recent years, increasingly more studies have used the dietary inflammatory index (DII) to evaluate the inflammatory effects of diet. DII can assess the influence of various foods or food components on inflammatory biomarkers based on a large number of studies [37]. It provides an estimate of the inflammatory potential of the overall diet [37]. Many studies have confirmed the strong correlation between DII and common inflammatory biomarkers [38]. DII also offers a new method for evaluating dietary association with disease progression [39,40,41]. The result of a prospective cohort study based on 1743 people shows that the DII scores in adults are positively correlated with depressive symptoms [39]. That is, the higher the DII score, the higher the risk of an adult developing depressive symptoms. Similarly, another prospective study in 2022 found that the rise in the DII increases the risk of depression [42].

To reduce the interruption rate of exclusive breastfeeding within 6 months in China, it is essential to ensure a reasonable diet for exclusive breastfeeding mothers, and reduce the incidence and impact of depression. The aim of this study was to evaluate the dietary quality by assessing the DII scores, exploring whether exclusive breastfeeding mothers’ diet contributes to an inflammatory environment in the body, which consequently affected the occurrence of depression. This study can serve as a reference for specific recommendations for improving the dietary structure of nursing mothers.

## 2. Materials and Methods

### 2.1. Ethical Approvals

The Ethics Review Committee of XiangYa School of Public Health, Central South University approved this cross-sectional study (No. XYGW-2020-51). Participants were required to sign an informed consent form prior to the investigation, and all data were kept strictly confidential.

### 2.2. Sampling

The study was conducted in Changsha, China, from July to October 2020. This is a cross-sectional study, and the study site was the Maternal and Child Health Hospital in Changsha, which has a relatively complete facility and system of child health care services.

In China, doctors usually recommend that parents bring their newborns to the hospital’s child health department for routine physical examinations at 42 days, 3 months, and 6 months after birth. The child health department of the hospital has a large number of infants seen daily and a wide distribution of population sources, which can better represent the basic situation of breastfeeding mothers in Changsha.

In this study, researchers screened women who brought their infants between 0 and 6 months of age to the child health department for regular checkups according to strict inclusion and exclusion criteria to obtain eligible study subjects.

### 2.3. Study Subjects

#### 2.3.1. Sample Size

The sample size was calculated using the PASS software package (NCSS, LLC, Kaysville, UT, USA). According to similar findings, women who exclusively breastfed within 6 months of birth had a 27.3% prevalence of depression [43], assuming an allowable error equal to 20% of prevalence. The required sample size was 266 (N = 266). Considering the 10% non-response rate, 293 women were ultimately required. In this study, 301 healthy women who exclusively breastfed within 6 months were included.

#### 2.3.2. Inclusion and Exclusion Criteria

The inclusion criteria were (1) physical health; (2) exclusive breastfeeding; (3) breastfeeding for less than or equal to 6 months; (4) normal comprehension and expression skills and willingness to participate in the study.

The exclusion criteria were (1) local residence for 3 months or more; (2) missing dietary data.

### 2.4. Data Collection

#### Questionnaires

All the data were collected by the investigators who received uniform training. After completion of the questionnaire, the investigator checked for any omissions or logical mistakes. The questionnaire consisted of three sections:

(1) Basic information on breastfeeding mothers: This section includes the demographic characteristics of the breastfeeding mother, the birth of the breastfeeding mother, and the demographic characteristics of the infant.

(2) Food Frequency Questionnaire (FFQ): A semi-quantitative FFQ was used to assess the dietary intake of breastfeeding mothers. The FFQ in this study was formulated according to “The 2016 Chinese Balanced Dietary Pagoda” [44]. The food types of FFQ were divided into 9 categories, namely grains, potatoes, vegetables, fruits, meat, fish and shrimp, eggs, milk, legumes, nuts, and fats, which could be subdivided into 45 entries. Frequency options included “never”; “less than or equal to 1 per month”; “2~3 per month”; “1 per week”; “2~3 per week”; “4~5 per week”; “1 per day”; “2 per day”; “less than or equal to 3 per day”.

(3) The Chinese version of the Edinburgh Postnatal Depression Scale (EPDS) [45]: The EPDS was used to assess the PPD status of breastfeeding mothers. The scale is composed of 10 items that address mindfulness, pleasure, self-blame, depression, fear, insomnia, coping skills, sadness, crying, and self-injury. Each item response is divided into 4 levels, reflecting the different severity of symptoms, from “never” to “always” with a score of 0 to 3 respectively (with 1 and 2 items being reverse scored). The scores of the 10 items were summed to obtain the total individual score, which ranged from 0 to 30. We used 9 as the cut-off point, with a total score of less than 9 indicating no PPD, 9–12 indicating mild PPD, and 13–30 indicating PPD.

(4) Pittsburgh Sleep Quality Index (PSQI) [46]: The scale consisted of 19 self-rating items and 5 other rating items, among which the 19th self-rating item and 5 other rating items do not participate in scoring. All the items were divided into seven dimensions, subjective sleep quality; sleep time; sleep latency; sleep efficiency; sleep disorders; hypnotic drugs and daytime function. The 18 self-assessment items involved in the scoring were composed of 7 components, and each component was scored according to the 0~3 level. Total PSQI scores were computed as the sum of the seven components from 0 to 21, with a higher score indicating poorer sleep quality. A PSQI score of 0 to 5 indicated satisfactory sleep quality, 6 to 12 indicated good sleep quality, 13 to 15 indicated moderate sleep quality, and 16 to 21 indicated poor sleep quality.

(5) Perceived Social Support Scale (PSSS): The PSSS was used to measure the perceived degree of social support from family, friends, and others [47]. The scale was made up of 12 self-assessments, and the responses are classified into 7 categories: strongly disagree; disagree; slightly disagree; neutral; slightly agree; consent; and strongly agree. Social support was measured by the overall score, ranging from 12 to 36 for low sense of social support, from 37 to 60 for moderate sense of social support, and from 61 to 84 for high sense of social support. The higher the overall score, the greater the sense of social support.

### 2.5. Statistical Analysis

#### 2.5.1. Dietary Inflammatory Index Calculation

This study ranked maternal dietary inflammatory potential and quality with DII, the development of which relied on 1943 previously published papers [37]. All these studies evaluated the dietary inflammatory potential of individuals based on the combined effects of 45 food parameters on six inflammatory biomarkers (IL-1β, IL-4, IL-6, IL-10, tumor necrosis factor-α, CRP) [48]. In previous studies, the construction and validation of DII have been described in detail [37]. The specific formulas are as follows [42]:Z score = (daily intake of this kind of dietary ingredient or nutrient − this kind of dietary ingredient or the global average per capita daily intake of nutrients)/the SD of the global average per capita daily intake of this dietary ingredient or nutrient
Z score1 = Z score → (converted to a percentile score) × 2 − 1
DII = ∑ Zscore1 × the inflammatory effect score of each dietary component

In this study, the matrix operation method of MATLAB R2019a (Mathworks, Inc., Natick, MA, USA.) was used to multiply the Z value of each nutrient/food center by its inflammatory effect score to obtain the DII score of each individual nutrient/food.

Compared with the traditional 45 nutrients, only 21 nutrients and foods were included in this study to obtain the final DII, It includes 6 pro-inflammatory components (energy, protein, carbohydrates, cholesterol, fat, saturated fatty acids) and 15 anti-inflammatory components (dietary fiber, monounsaturated fatty acids, polyunsaturated fatty acids, carotene, vitamin A, thiamine, riboflavin, vitamin C, vitamin E, folic acid, niacin, iron, magnesium, selenium, zinc). The reasons are as follows: (1) in this study, the latest version of Chinese food composition table was used to find the nutrient content of food, but some nutrients or food could not be obtained in this table, such as flavonols, flavanones, anthocyanins, and flavan-3-ols; (2) the dietary diversity of Chinese breastfeeding mothers is limited by traditional customs; some foods such as coffee, thyme/oregano, rosemary, and alcohol are consumed at low or no levels.

A lower DII score indicates an anti-inflammatory diet, while a higher DII score indicates a pro-inflammatory diet. Due to the limited sample size of the study population, if the study subjects were grouped based on DII by the quartile method, the number of people in each group is too small, and the difference between the groups is difficult to be found. Therefore, according to the DII score of the subjects, the population was divided into three groups according to the tertile method.

Three DII groups were defined based on the DII scores. The DII scores were ranked from lowest to highest. Those who entered the top 33.3% were assigned to group Q_1_, the lowest scoring group; those who entered between 33.3% and 66.7% were assigned to group Q_2_; and those above 66.7% were assigned to group Q_3_, the highest scoring group.

According to the Dietary Guidelines for Pregnant Women and breastfeeding Mothers in China (2022) [49], for the extreme values of energy intake, 8 people with daily energy intake ≥ 3500 kcal were excluded, leaving 293 people being included in the final analysis.

#### 2.5.2. Data Analysis

Statistical description, comparison and regression analysis were performed with SPSS25.0 (IBM Corp., Armonk, NY, USA). The DII score and nutritional intake of the breastfeeding mothers was described by mean ± standard deviation. The quantitative data were transformed into qualitative data and expressed as a percentage, including age, spouse, household income, children, postpartum caregiver, PSSS score, PSQI score, and DII score. The prevalence rate of PPD in each group was compared by chi-square method. With PPD as the dependent variable and DII as the independent variable, a logistic regression equation was constructed to analyze the association between DII and PPD after adjusting by confounding factors.

## 3. Results

### 3.1. General Information of the Participants

A total of 293 study participants were included in the analysis, with a median EPDS score of 10. Figure 1a shows that study participants were divided into six groups based on EPDS scores, with 117 participants (39.9%) in the lowest scoring group (5~8). As the EPDS score increased, the number of participants in each group decreased. Table 1 describes the demographic characteristics, social support levels, sleep levels and DII levels of the breastfeeding women grouped according to PPD status. The results show that the majority of the participants were 26 to 35 years old (231/293, 78.80%), and half of the families lived in urban area (151/293, 51.5%). In the comparison of the PPD rate between groups, we found that participants with junior high school education and below, only children, only one postpartum caregiver, and low social support level were more likely to suffer from PPD. Participants who were unemployed and had good sleep quality were less likely to be depressed.

### 3.2. Dietary Inflammatory Index and Its Influencing Factors

The mean DII score was 2.32 ± 1.08, ranging from −1.66 to 4.19. The median DII score was 2.51, and Figure 1b suggests that more than 50% (196) of the participants scored between 2 and 4. The DII scores were grouped by tertile, with a DII score below 2.06 in Q_1_, DII in Q_2_ from 2.06 to 2.89, and DII in Q_3_ over 2.89.

Table 2 describes the demographic characteristics, social support, and sleep levels of breastfeeding mothers grouped classified by DII scores. Breastfeeding mothers with Han Chinese spouses, more than three postpartum caregivers, and moderate to low levels of social support were more inclined to the maximum pro-inflammatory diet. Appendix A shows that higher intakes of pro-inflammatory nutrients (dietary fat, cholesterol, saturated fatty acids, etc.) were associated with higher DII scores. The higher the intake of anti-inflammatory nutrients (dietary fiber, vitamin A, vitamin B1, vitamin B2, vitamin C, vitamin E, iron, zinc, monounsaturated and polyunsaturated fatty acids, folic acid, etc.), the lower the DII score. Notably, the rise in DII levels was associated with increased levels of three pro-inflammatory nutrients (energy, protein, and sugar).

### 3.3. Factors Related to PPD

With the present depression as the dependent variable, the relationship between different DII groups (the DII score greater than 2.89 (pro-inflammatory diet) as the control) and depression was analyzed (Table 3). The results showed that a DII score less than 2.89 was a protective factor for postpartum depression, and its significance still existed after a series of factors were adjusted. In model 1, Q_1_ compared to Q_3_ (OR: 0.50, 95% CI: 0.27, 0.94), and Q_2_ compared to Q_3_ (OR: 0.406, 95% CI: 0.22, 0.77). In model 2, Q_1_ compared to Q_3_ (OR: 0.47, 95% CI: 0.24, 0.93), and Q_2_ compared to Q_3_ (OR: 0.38, 95% CI: 0.19, 0.74).

Appendix A in the appendix presents odds ratios and 95% confidence intervals for depression between groups of influencing factors. Model 2, which adjusted a series of influencing factors, shows that breastfeeding women with one caregiver and low or medium social support score were more likely to have depressive symptoms than those with more than or equal to three caregivers and high social support score (OR: 3.93, 95% CI: 1.46, 10.55; OR: 1.89, 95% CI: 1.05, 3.43). Results show that lower PSQI scores were associated with better sleep quality, and that higher sleep quality may be a protective factor for depressive symptoms (OR: 0.47, 95% CI: 0.27, 0.84). In Model 1, the significance of risk factors for PPD such as middle school education or below and being an only child still existed in model 2 (OR: 10.84, 95% CI: 1.00, 117.56; OR: 2.10, 95%CI: 1.74, 3.74). In both model 1 and model 2, non-employment was a protective factor for depressive symptoms. (OR: 0.49, 95% CI: 2.29, 0.85; OR: 0.48, 95% CI: 0.27, 0.86).

## 4. Discussion

In the present study, the relation between DII and PPD was investigated in a cross-sectional study of Chinese breastfeeding women. In this study, we calculated DII scores from dietary data collected by FFQ, assessed postpartum depression in breastfeeding mothers using the EPDS scale, and finally explored the association between DII and depression in breastfeeding mothers during the 6 month postpartum. The results of the study showed that when breastfeeding women moved from the most pro-inflammatory diet (highest DII tertile, Q_3_) to the most anti-inflammatory diet (lowest DII tertile, Q_1_), the risk of PPD decreased.

In this study, 60.1% participants experienced PPD. Similar studies using EPDS (≥9 or 10 points) to assess postpartum depression showed that the PPD rate was 30.0~38.0% [50,51,52]. Using EPDS (≥12 or 13 points) as the cut-off value, the depression rate of breastfeeding mothers after delivery was 11.8~27.3% [8,43,53]. The reasons for the high rate of PPD in this study may be as follows. (1) There is a vast difference in the incidence of PPD between developing countries and developed countries. According to a meta-analysis by M.N. Norhayati et al., the incidence rate of postpartum depression in developing countries was 1.9~82.1%, higher than the 5.2~74.0% in developed countries [54]. (2) This study adopted 9 points as the cut-off point for EPDS assessment of depression, which is more sensitive [55]. (3) In this study, 38.2% of the participants were exclusively breastfed for 1 month, while only 21.9% were breastfed for 4 to 6 months. It has been suggested that the rate of postpartum depression decreases with time, reaching a low level by 6 months after delivery. This may also account for the high incidence of PPD in mothers (exclusive breastfeeding within 6 months) in this study, as compared with those who had been breastfed exclusively for 6 months in other studies. (4) This study was conducted at the beginning of the COVID-19 pandemic, and the fear of the pandemic may increase the anxiety and depression of lactating mothers [56]. A 2022 meta-analysis also showed that the combined prevalence of postpartum depression during the COVID-19 pandemic was 34% (95% CI: 21~46%), much higher than that in previous studies during non-pandemic periods [57]. (5) Differences in the study population may be attributable to the high rate. The subjects of this study were exclusively breastfeeding mothers, while the association between breastfeeding and depression is unclear. Studies have shown that incorrect breastfeeding or feeding difficulties may increase maternal depression [58,59,60]. Although the detection rate of depression in this study is rather high, our result is consistent with reports in Asian countries which indicated the prevalence of PPD ranged from 3.5 to 63.3% [61], which is still in a reasonable range. In the analysis, we adopted a multivariate analysis strategy to adjust the occupational levels, sleep quality, and social support levels that may affect the incidence of depression during such a special period of the pandemic, and did not find any serious impact on the analysis of the relationship between the DII and PPD in exclusively breastfeeding mothers. Therefore, we believe that the high detection rate of this study did not have an impact on the results.

The theory that postpartum depression in breastfeeding mothers may increase the risk of breastfeeding failure is due to the hypothesis that depressed mothers are less confident in their ability to breastfeed, and therefore less willing to continue breastfeeding than mothers without depressive symptoms [62]. In addition, the mental status of breastfeeding mothers can directly stimulate or inhibit their cerebral cortex, thus stimulating or inhibiting the release of prolactin and oxytocin, and it can also affect the lactation and milk ejection process through the neuronal endocrine mechanism, which in turn directly affects the breastfeeding of infants from 0 to 6 months.

In this study, the average DII score is 2.32, suggesting that the study population is on a pro-inflammatory diet, and it is consistent with a previous average DII score of 1.81 for breastfeeding mothers in China. However, a study on postpartum diet and bone mineral density showed an average DII score of −0.039 [63]. Moreover, in another study of DII in adult women, the average score was 1.1 [64]. The different results may be due to the differences in countries, regions, and ethnic groups. In addition, postpartum women are in a special physiological state—the puerperium. The puerperium is the first stage of lactation. It is the period of about six weeks after childbirth during which the mother’s reproductive organs return to their original pre-pregnant condition [65]. Chinese people attach great importance to postpartum health care [66]. With a history of more than 2000 years, “zuoyuezi” or “doing the month” is a Chinese postpartum tradition in which the mother and newborn(s) are surrounded together by their families and community members [65,67]. Puerperal women are supposed to follow the traditional dietary habit, eating more warm foods or yang foods (they bring heat or warmth to the body) such as animal foods and ginger, and eating fewer cool foods or yin foods (they have a cooling effect on the body) such as vegetables and fruits [68,69,70,71]. Presently, most Chinese puerperal women still follow the traditional dietary habit to some extent, which could explain the pro-inflammatory diet of postpartum breastfeeding mothers in China [72].

This study found that lowering the DII scores can reduce the risk of PPD. Compared to the group (Q_3_) with a DII score above 2.89, lowering the score to 2.06 or 2.89 (Q_1_, Q_2_) can reduce the risk of PPD by almost 50% (OR 0.47, 0.38, respectively). Moreover, this protective effect remained after the occupational levels, sleep quality, and social support levels of the subjects were adjusted. This finding of association between a pro-inflammatory diet and PPD was consistent with the findings of earlier studies. A longitudinal study in Australia reported that a pro-inflammatory diet was associated with a higher possibility of depression among middle-aged women in Australia [64]. Another study of adults showed that there is a more than twofold increase in the risk of depression for each unit increase in the DII score [39]. According to the data from the Korea National Health and Nutrition Survey, people in the highest quartile of the DII have a 1.44 times higher risk of depression than those in the lowest quartile in Seoul [73].

The pro-inflammatory diet is associated with pro-inflammatory cytokines, which can affect the content and conduction of neurotransmitters, causing depression to occur [24]. This effect was also found in the male population and in other diseases. A study in China showed that higher DII was associated with an increased risk of depression among participants with chronic diseases and comorbidities, especially those male participants under the age of 60 [41]. There are a small number of studies, however, that have yielded the opposite result. In a prospective study, it was observed that DII was not associated with depressive symptoms in the entire population, and a mildly significant positive correlation was observed in the female population [74]. Yet, a recent meta-analysis has shown that a lower intake of pro-inflammatory diets is associated with a lower risk of depression [75].

The dietary patterns of Chinese women during confinement and lactation often undergo a major shift due to traditional customs [76]. During this period, breastfeeding mothers are required to consume a large amount of nourishing broth and reduce the intake of cool fruits and vegetables to ensure that the mother can maintain the balance of yin and yang in the body after childbirth as well as provide the baby with sufficient breast milk [67]. However, the change in diet also brings a risk of inflammation to the body [77], which can easily cause an unhealthy mood. In addition, compared with the general population, breastfeeding mothers usually have a more complicated mechanism for regulating mood-related hormones [77,78]. Significant fluctuations in hormone levels in breastfeeding women may attenuate or enhance the inflammatory effects of the diet, leading to different results. Therefore, based on existing studies on inflammatory diets associated with depression and the results of this study, further prospective cohort studies are needed to confirm the evidence. Given the dietary characteristics and high DII scores of breastfeeding mothers in China, we can still cautiously advise that Chinese breastfeeding mothers should eat more anti-inflammatory foods or less pro-inflammatory foods to reduce depression or unhealthy emotions in order to maintain breastfeeding.

The strengths of this study are that the study has a large enough research sample size and adjusts for a range of possible confounding factors. In addition, it is understood to be an innovative exploration to study the relationship between DII and postpartum depression in exclusively breastfeeding women in China. However, we should acknowledge that the study had several limitations. Firstly, the practice of enrolling study subjects in only one maternity hospital limits the extrapolation of conclusions. However, the diet of Chinese lactating mothers is relatively homogenous, and our results can still reflect the situation of Chinese lactating mothers and the association between DII and depression to a certain extent. Secondly, although the FFQ used in this study was modified on the basis of the China Nutrition and Health Surveillance Questionnaire, it was not revalidated, which may affect the validity and reliability of the revised FFQ questionnaire. In addition, the subjects will recall their dietary intake in the past month, which may lead to certain recall bias. In addition, this study is a cross-sectional study, which only investigated the relationship between DII and depression at one time point and cannot infer the causal relationship between DII score and postpartum depression. At the same time, the regulation of inflammatory state by diet is a long and subtle process. Prospective studies are needed to confirm our findings. However, as the first study to focus on the relationship between DII score and depression in exclusive breastfeeding lactating women in China, our results still provide a meaningful reference for better formulation and guidance of lactating diets, thereby improving and maintaining exclusive breastfeeding rates.

## 5. Conclusions

An anti-inflammatory diet (DII < 2.89) was a protective factor for postpartum depression. The risk of postpartum depression in exclusive breastfeeding mothers decreased by nearly 50% (OR = 0.47) when they switched from the most pro-inflammatory diet (DII highest quartile, Q_3_) to the most anti-inflammatory diet (DII highest quartile, Q_1_). This suggests that breastfeeding women in China who reduce the inflammatory scores in their diet by adjusting their current dietary patterns seem to have great potential in preventing postpartum depression and thus reducing the risk of interrupting breastfeeding.

## Figures and Tables

**Figure 1 nutrients-14-05006-f001:**
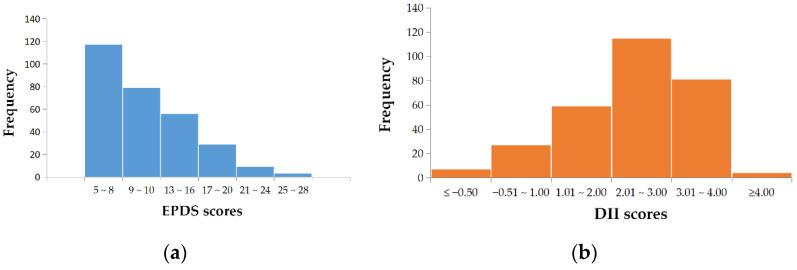
Distribution of EPDS and DII scores. (**a**) Distribution of EPDS scores. (**b**) Distribution of DII scores. (EPDS, Edinburgh Depression Scale. DII, dietary inflammatory index).

**Table 1 nutrients-14-05006-t001:** Distribution of PPD across sociodemographic, social support, sleep levels, and DII levels (*n*, %).

Variables	Total (n)	The Status of PPD		*p* Value ^1^
		Non-PPD	PPD	
Age groups (years)				0.306
≤25	42	18, 42.9	24, 57.1	
26~35	231	88, 38.1	143, 61.9	
36~45	20	11, 55.0	9, 45.0	
Ethnic groups				0.996
Han	278	111, 39.9	167, 60.1	
Minority	15	6, 40.0	9, 60.0	
Educational levels				0.021 **
Junior high school and below	10	1, 10.0	9, 90.0	
Specialized degree	126	42, 33.3	84, 66.7	
Bachelor’s degree	126	58, 46.0	68, 54.0	
Postgraduate and above	31	16, 51.6	15, 48.4	
Occupational levels				0.069 *
Employed	191	69, 36.1	122, 63.9	
Unemployed	102	48, 47.1	54, 52.9	
Region				0.757
Suburb	142	58, 40.8	84, 59.2	
Urban	151	59, 39.1	92, 60.9	
Spouse’s age groups (years)				0.859
≤25	26	10, 38.5	16, 61.5	
26~35	215	87, 40.5	128, 59.5	
36~45	52	20, 38.5	32, 61.5	
Spouse’s ethnic groups				0.996
Han	278	111, 39.9	167, 60.1	
Minority	15	6, 40.0	9, 60.0	
Spouse’s educational Levels ^#^				0.224
Junior high school and below	10	6, 60.0	4, 40.0	
Specialized degree	117	42, 35.9	75, 64.1	
Bachelor’s degree	129	50, 38.8	79, 61.2	
Postgraduate and above	36	19, 52.8	17, 47.2	
Household income levels (RMB per month)				0.416
≤6000	41	14, 34.1	27, 65.9	
6000~10,000	132	58, 43.9	74, 56.1	
≥10,000	120	45, 37.5	75, 62.5	
Baby’s age groups (months)				0.213
≤1	112	51, 45.5	61, 54.5	
2~3	117	40, 34.2	77, 65.8	
≥4	64	26, 40.6	38, 59.4	
Number of children				0.054 *
1	180	64, 35.6	116, 64.4	
≥2	113	53, 46.9	60, 53.1	
Number of caregivers ^#^				0.016 **
1	52	13, 25.0	39, 75.0	
2	197	81, 41.1	116, 58.9	
≥3	43	23, 53.5	20, 46.5	
PSSS score levels				0.001 **
Low to medium social support	100	27, 27.0	73, 73.0	
High social support	193	90, 46.6	103, 53.4	
PSQI score levels ^#^				0.001 **
Good sleep quality	86	47, 54.7	39, 45.3	
Moderate or poor sleep quality	205	70, 34.1	135, 65.9	
DII groups				0.022 **
Q_1_	98	43, 43.9	55, 56.1	
Q_2_	98	46, 46.9	52, 63.1	
Q_3_	97	28, 28.9	69, 71.1	
Total	293	117, 39.9	176, 60.1	

PPD, postpartum depression; non-PPD, non-postpartum depression; PSSS, Perceived Social Support Scale; PSQI, Pittsburgh Sleep Quality Index; DII, dietary inflammatory index. ^#^ Missing data not included. ^1^ Chi-square test. * *p* < 0.1, ** *p* < 0.05.

**Table 2 nutrients-14-05006-t002:** Distribution of DII levels across sociodemographic, social support and sleep levels (*n*, %).

Variables	DII Tertiles			*p* Value ^1^
	Q_1_	Q_2_	Q_3_	
Age groups (years)				0.261
≤25	10, 23.8	17, 40.5	15, 35.7	
26~35	79, 34.2	73, 31.6	79, 34.2	
36~45	9, 45.0	8, 40.0	3, 15.0	
Ethnic groups				0.432
Han	91, 32.7	95, 34.2	92, 33.1	
Minority	7, 46.7	3, 20.0	5, 33.3	
Educational levels				0.298
Junior high school and below	0, 0.0	5, 50.0	5, 50.0	
Specialized degree	41, 32.5	42, 33.3	43, 34.1	
Bachelor’s degree	45, 35.7	40, 31.7	41, 32.5	
Postgraduate and above	12, 38.7	11, 35.5	8, 25.8	
Occupational levels				0.256
Employed	68, 35.6	66, 34.6	57, 29.8	
Unemployed	30, 29.4	32, 31.4	40, 39.2	
Region				0.256
Suburb	42, 29.6	47, 33.1	53, 37.3	
Urban	56, 37.1	51, 33.8	44, 29.1	
Spouse’s age groups (years)				0.539
≤25	8, 30.8	9, 34.6	9, 34.6	
26~35	71, 33.0	68, 31.6	76, 35.3	
36~45	19, 36.5	21, 40.4	12, 23.1	
Spouse’s ethnic groups				0.035 *
Han	89, 32.0	97, 34.9	92, 33.1	
Minority	9, 60.0	1, 6.7	5, 33.3	
Spouse’s educational Levels ^#^				0.383
Junior high school and below	5, 50.0	4, 40.0	1, 10.0	
Specialized degree	35, 29.9	40, 34.2	42, 35.9	
Bachelor’s degree	42, 32.6	46, 35.7	41, 31.8	
Postgraduate and above	16, 44.4	8, 22.2	12, 33.3	
Household income levels (RMB per month)				0.951
≤6000	12, 29.3	14, 34.1	15, 36.6	
6000~10,000	46, 34.8	45, 34.1	41, 31.1	
≥10,000	40, 33.3	39, 32.5	41, 34.2	
Baby’s age groups (months)				0.469
≤1	36, 32.1	36, 32.1	40, 35.7	
2~3	45, 38.5	36, 30.8	36, 30.8	
≥4	17, 26.6	26, 40.6	21, 32.8	
Number of children				0.897
1	59, 32.8	62, 34.4	59, 32.8	
≥2	39, 34.5	36, 31.9	38, 33.6	
Number of caregivers ^#^				0.004 *
1	14, 26.9	24, 46.2	14, 26.9	
2	78, 39.6	56, 28.4	63, 32.0	
≥3	6, 14.0	17, 39.5	20, 46.5	
PSSS score levels				0.044 *
Low to medium social support	26, 26.0	32, 32.0	42, 42.0	
High social support	72, 37.3	66, 34.2	55, 28.5	
PSQI score levels ^#^ Age groups (years)				0.783
Good sleep quality	29, 33.7	31, 36.0	26, 30.2	
Moderate or poor sleep quality	68, 33.2	67, 32.7	70, 34.1	
Total	98, 33.4	98, 33.4	97, 33.0	

PSSS, Perceived Social Support Scale; PSQI, Pittsburgh Sleep Quality Index; DII, dietary inflammatory index. ^#^ Missing data not included. ^1^ Chi-square test. * *p* < 0.05.

**Table 3 nutrients-14-05006-t003:** Binary logistic regression model of the association between DII score and PPD among exclusively breastfeeding women (*n* = 293).

	DII Tertiles (OR, 95% CI)			
	Q_1_	Q_2_	Q_3_	
	*n* = 98	*n* = 98	*n* = 97	*p* Value ^1^
Crude Model	0.52 (0.29, 0.94)	0.46 (0.25, 0.83)	1	0.024 *
Mode l	0.50 (0.27, 0.94)	0.41 (0.22, 0.77)	1	0.016 *
Mode 2	0.47 (0.24, 0.93)	0.38 (0.19, 0.74)	1	0.013 *

PPD, postpartum depression; DII, dietary inflammatory index. ^1^ Binary logistic regression. Crude Model: unadjusted for any covariate. Mode 1: adjusted age (≤25, 26~35, 36~45), educational levels (junior high school and below, specialized degree, bachelor’s degree, postgraduate and above), occupational levels (employed, unemployed), number of children (1, ≥2). Model 2: adjusted for age (≤25, 26~35, 36~45), educational levels (junior high school and below, specialized degree, bachelor’s degree, postgraduate and above), occupational levels (employed, unemployed), number of children (1, ≥2), number of postpartum caregivers (1, 2, ≥3), PSSS score level (low to medium social support, high social support) and PSQI score level (good sleep quality, moderate or poor sleep quality). * *p* < 0.05.

## Data Availability

The data that support the findings of this study are not publicly available, due to the data containing information that could compromise the participants’ privacy, but are available from the corresponding author upon reasonable request.

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
