# Peer review of "Relationship between Dietary Inflammatory Index and Postpartum Depression in Exclusively Breastfeeding Women"

_nutrients, 2022, doi:10.3390/nu14235006_

Round 1
Reviewer 1 Report
table 1 & 2
Total% /The status of PPD% or Total%/ DII Tertiles% were not made appropriately (Total% should be between each group% ?)
Reviewer 2 Report
Title: Relationship between Dietary Inflammatory Index and Postpartum
Depression in Exclusively Breastfeeding Women
Authors correlated the inflammatory index of the diet with the risk of postpartum depression.
The whole paper is presented in a clear way. As the main subject of the paper is postpartum depression would be beneficial to provide some more information about the EPDS. What were the scores or items that the authors mentioned in the paper? It would be easier to understand without referring to the appropiate reference position.
The bid advanced of the presented paper is statistics, its description, the presentation of the data, and proper usage. The tables are clean and easy to understand.
Improving some edition errors (for example 5 lines 218 and219 where repetitions make the line almost the same). Please check the whole manuscript to avoid this type of error.
Round 2
Reviewer 1 Report
Much improved